**Data Availability Statement:** All relevant data are within the paper.

**Funding:** The study was supported by the National Research Foundation (KR) NRF-2021R1A2C1011579. The funders had no role in

# Multilocus genotyping of *Giardia duodenalis* in pre-weaned calves with diarrhea in the Republic of Korea

**Yu-Jin Park**[1], **Hyung-Chul Cho**[1], **Dong-Hun Jang**[1], **Jinho Park**[2], **Kyoung-Seong Choi**📧[1]*

**1** Department of Animal Science and Biotechnology, College of Ecology and Environmental Science, Kyungpook National University, Sangju, Republic of Korea, **2** College of Veterinary Medicine, Jeonbuk National University, Iksan, Republic of Korea

* kschoi3@knu.ac.kr

## Abstract

*Giardia duodenalis* is a protozoan parasite that infects humans, companion animals, livestock, and wildlife. Infections in cattle caused by this parasite are often asymptomatic, but such infections can cause diarrhea, reduced weight gain, and ill-thrift in young calves. Although *G. duodenalis* causes diarrhea in calves, only a few studies have been conducted on calves in the Republic of Korea (ROK). Here, we aimed to determine the prevalence and distribution of *G. duodenalis* assemblages in pre-weaned calves with diarrhea in the ROK, identify the association between the occurrence of *G. duodenalis* and the age of calf, and perform molecular characterization of *G. duodenalis*. We collected 455 fecal samples from pre-weaned native Korean calves (≤60 days old) with diarrhea in four different regions. *G. duodenalis* was detected using nested PCR targeting the beta-giardin (*bg*) gene, and positive samples were further genotyped for the glutamate dehydrogenase (*gdh*) and triosephosphate isomerase (*tpi*) genes. The overall prevalence of *G. duodenalis* in calves with diarrhea was 4.4% (20/455) based on the analysis of *bg*. The highest prevalence was observed in calves aged 11−30 days (7.5%; 95% confidence interval: 3.7%–11.3%), whereas the lowest prevalence was observed in neonatal calves. From the 20 samples that were positive for *bg*, 16, 5, and 6 sequences were obtained following genotyping of *bg*, *gdh*, and *tpi*, respectively. Sequencing analysis of the *bg* gene revealed the presence of assemblage E ($n = 15$) and sub-assemblage AI ($n = 1$) in the samples. Moreover, we detected mixed infections with assemblages E and A in two calves for the first time. Among the sequences obtained herein, two new subtypes of assemblage E were detected in *gdh* and *tpi* sequences each. The results suggest that *G. duodenalis* is an infectious agent causing diarrhea in calves, and pre-weaned calves are at a higher risk of infection than neonatal calves. Multilocus genotyping should be performed to confirm the presence of potentially zoonotic genotypes. These results highlight the importance of cattle as a source of zoonotic transmission of *G. duodenalis* to humans.

study design, data collection and analysis, decision to publish, or preparation of the manuscript.

**Competing interests:** The authors have declared that no competing interests exist.

## Introduction

*Giardia duodenalis* is a zoonotic enteric protozoan parasite that can infect a wide range of animals, including humans [1]. *G. duodenalis* infection in humans and animals causes watery diarrhea, malabsorption, and weight loss and may result in death in extreme cases [2, 3]. This parasite is mainly transmitted via the fecal–oral route or via the ingestion of cyst-contaminated food or water; moreover, it can be transmitted through direct contact with infected animals [4]. *G. duodenalis* is currently classified into eight host-specific assemblages (A−H) based on molecular characterization. Assemblages A and B are zoonotic and have broad host ranges, which include livestock, companion animals, and humans, whereas assemblages C−H have the following specific hosts: C and D in canids, E in livestock, F in felids, G in rodents, and H in marine mammals [3, 5]. Assemblage E is the dominant genotype in hoofed farm animals, such as pigs, sheep, goats, and cattle, and recent studies have also identified this assemblage in humans in Australia, Brazil, and Egypt [6–8], highlighting potential zoonotic transmission.

*Giardia duodenalis* is an important pathogen in young calves and causes diarrhea and ill-thrift, leading to enormous economic loss in the livestock industry [9–12]. Its worldwide prevalence ranges from 5% to 55.4%, as determined using molecular analysis [13]. Meanwhile, its prevalence in farms varies between 45% and 100% [11]. Notably, cattle are known to be potential reservoir of human infections, increasing the public health significance of *G. duodenalis* infection in cattle, which excrete high numbers of cysts into the environment. Assemblages A, B, and E have been identified in cattle to date [1, 14, 15], with assemblage E being the most prevalent [16]. Assemblages A and B are further divided into three (AI, AII, and AIII) and two (BIII and BIV) sub-assemblages, respectively, that are genetically similar and closely associated with each other [17]. A recent study reported the presence of 34 subtypes within assemblage E based on the analysis of the beta-giardin (*bg*) gene [18]. In particular, assemblage E is characterized by a high degree of genetic variation [15, 19, 20]. However, studies identifying the subtypes of assemblage E remain insufficient [19, 21–23]. In addition, the phylogenetic tree-based nomenclature for the subtypes of assemblage E is ambiguous.

For the detection and molecular characterization of *G. duodenalis*, several studies have involved the targeting of various genes, such as the small subunit rRNA (SSU rRNA), *bg*, glutamate dehydrogenase (*gdh*), and triosephosphate isomerase (*tpi*) genes [3, 12]. These genes can be used to differentiate between assemblages involved in mixed infections [3, 24]. Although *G. duodenalis* is known to cause diarrhea in calves, only few studies have been conducted in the Republic of Korean (ROK) regarding its prevalence, relationship with diarrhea, genotyping, and intra-assemblage variation [25–28]. Therefore, the present study aimed to determine the prevalence and distribution of *G. duodenalis* assemblages in calves with diarrhea in the ROK, identify the association between the occurrence of *G. duodenalis* infection and the age of calf, and assess the genotypes within assemblage E based on the analysis of three genes.

## Materials and methods

### Ethics statement

All animal procedures were conducted according to ethical guidelines for the use of animal samples and were approved by the Jeonbuk National University (Institutional Animal Care and Use Committee Decision No. CBNU 2020–052). All procedures and possible consequences were explained to the managers of the surveyed farm, and written consent was obtained.

### Sample collection

From August 2019 to February 2022, fecal samples were collected from 455 individual pre-weaned native Korean calves aged ≤60 days. The fecal samples were divided into three groups

according to the age of the calves: 1−10 days ($n = 219$), 11−30 days ($n = 186$), and 31−60 days ($n = 50$). Fecal samples were collected from four different regions (Gyeonggi, Jeonbuk, Gyeongbuk, and Gyeongnam provinces) in the ROK. These fecal samples were directly collected from the rectum of each diarrheic calf by a veterinarian using sterile disposal latex glove into a specimen cup, labeled, placed onto ice, and transported to the laboratory in Kyungpook National University as soon as possible. For each animal, age, gender, geographical location, sampling date, and fecal consistency were recorded. The number of samples collected in different seasons were as follows: spring ($n = 98$), summer ($n = 135$), fall ($n = 150$), and winter ($n = 72$). Each animal was sampled only once during the study period. Before DNA extraction, all samples were stored at −20˚C.

## DNA extraction and polymerase chain reaction (PCR)

Genomic DNA was extracted from 100−200 mg of each fecal sample using the AccuPrep® Stool DNA Extraction Kit (Bioneer, Daejeon, ROK) in accordance with the manufacturer's instructions. These DNA extracts were frozen at −20˚C until further analysis. The samples were first screened for *G. duodenalis* using nested PCR targeting the *bg* gene, and the positive samples were further analyzed for the *gdh*, and *tpi* genes. Information regarding each primer is listed in Table 1. *bg* was amplified under different annealing temperatures in the primary and secondary PCRs as follows: 94˚C for 15 min; followed by 35 cycles of 95˚C for 30 s, 65˚C and 55˚C for 30 s for primary and secondary PCRs, respectively, and 72˚C for 60 s; and a final extension at 72˚C for 7 min. The PCR conditions for *gdh* were as follows: 94˚C for 2 min; followed by 35 cycles of 95˚C for 30 s, 58˚C for 30 s, and 72˚C for 60 s; and a final extension of 72˚C for 7 min. *tpi* was amplified under the following conditions: 94˚C for 5 min; followed by 35 cycles of 94˚C for 45 s, 50˚C for 45 s, and 72˚C for 60 s; and a final extension at 72˚C for 10 min. Distilled water was used as a negative control in all PCRs. Amplified PCR products were visualized on 1.5% agarose gels stained with ethidium bromide.

## Sequencing and phylogenetic analysis

All secondary PCR amplicons were purified using an AccuPrep® PCR Purification Kit (Bioneer, Daejeon, ROK) according to the manufacturer's instructions. The purified amplicons were used for direct sequencing (Macrogen, Daejeon, ROK), and the resulting sequences were aligned using BioEdit software. The aligned sequences were compared with the reference

**Table 1. Primers used to amplify *bg*, *gdh*, and *tpi*.**

| Gene | Primer | Sequence (5′–3′) | Annealing temp. (˚C) | Amplicon size (bp) |
|---|---|---|---|---|
| *bg* | G7F | AAGCCCGACGACCTCACCCGCAGTGC | 65 | 753 |
| | G759R | GAGGCCGCCCTGGATCTTCGAGACGAC | | |
| | G7nF | GAACGAACGAGATCGAGGTCCG | 55 | 511 |
| | G759nR | CTCGACGAGCTTCGTGTT | | |
| *gdh* | GDH1 | TTCCGTRTYCAGTACAACTC | 55 | 754 |
| | GDH2 | ACCTCGTTCTGRGTGGCGCA | | |
| | GDH3 | ATGACYGAGCTYCAGAGGCACGT | | 520 |
| | GDH4 | GTGGCGCARGGCATGATGCA | | |
| *tpi* | AL3543 | AAATIATGCCTGCTCGTCG | 50 | 605 |
| | AL3546 | CAAACCTTITCCGCAAACC | | |
| | AL3544 | CCCTTCATCGGIGGTAACTT | | 532 |
| | AL3545 | GTGGCCACCACICCCGTGCC | | |

sequences from GenBank to identify *G. duodenalis* assemblages. Samples that yielded *bg*, *tpi*, and *gdh* amplicons were further analyzed via multilocus genotyping (MLG) to reveal their genetic diversity. If one or more variants were detected using MLG, then the sequence was referred to as a novel sequence subtype [16, 18–20, 29].

Phylogenetic analysis based on each gene was conducted using the maximum-likelihood method implemented in MEGA11 using the best substitution model. Bootstrap values were calculated by analyzing 1,000 replicates to evaluate the reliability of clusters. The models used in this study were Tamura-Nei 93 (TN93) for *bg* and *tpi*, and Tamura-3-parameter (T92) for *gdh*. Novel sequence subtypes were identified, and these sequences were named according to previous studies [14, 19].

## Statistical analysis

Data were analyzed using the SPSS package version 26 (IBM Corp.; Armonk, NY, USA). Chi-square test was used to determine the association between the prevalence of *G. duodenalis* and each age group (i.e., 1–10, 11–30, or 31–60 days). $P < 0.05$ was considered statistically significant.

## Nucleotide sequence accession numbers

The nucleotide sequences obtained in the present study have been deposited in the GenBank database under the accession numbers: ON677352–ON677367 for *bg*, OP271725–OP271729 for *gdh*, and OP271719–OP271724 for *tpi*.

## Results

### Prevalence of *G. duodenalis*

The overall prevalence of *G. duodenalis* in pre-weaned calves with diarrhea was 4.4% (20/455) based on the analysis of *bg*. According to the regions, the highest and lowest prevalence of *G. duodenalis* was observed in Gyeongnam (6.6%) and Gyeonggi (2.5%) provinces, respectively (Table 2). However, this regional difference was not statistically significant. Based on the age group of calves, *G. duodenalis* was the most prevalent in calves aged 11–30 days (7.5%; 95% confidence interval [CI]: 3.7%–11.3%), followed by those aged 31–60 days (6%; 95% CI: 0%–12.6%) and 1–10 days (1.4%; 95% CI: 0%–2.9%) (Table 3). The *G. duodenalis* infection rate in calves aged 11–60 days was significantly higher than that in neonatal calves aged 1–10 days ($\chi^2$ = 9.417, $P = 0.009$). *G. duodenalis* was found to be associated with diarrhea in pre-weaned calves but not in neonatal calves.

**Table 2. Prevalence and subtypes of *G. duodenalis* among pre-weaned calves according to region.**

| Region | No. of positive samples | Prevalence (95% CI) | No. of subtypes [a] (No. of sequenced sample) | MLG |
|---|---|---|---|---|
| Gyeonggi (*n* = 40) | 1 | 2.5% (0.0%–7.3%) | AI (0), E1 (0), E3 (1), E5 (0), E11 (0) | |
| Jeonbuk (*n* = 273) | 10 | 3.7% (1.4%–5.9%) | AI (0), E1 (3), E3 (3), E5 (3), E11 (0) | MLG-E1 (1), MLG-E2 (1) |
| Gyeongnam (*n* = 91) | 6 | 6.6% (1.5%–11.7%) | AI (1), E1 (0), E3 (1), E5 (0), E11 (4) | |
| Gyeongbuk (*n* = 51) | 3 | 5.9% (0.0%–12.3%) | – | |
| Total (*n* = 455) | 20 | 4.4% (2.5%–6.3%) | AI (1), E1 (3), E3 (5), E5 (3), E11 (4) | E1 (1), E2 (1) |

"a": subtype based on the *bg* locus.

"–": none of the samples sequenced.

**Table 3. Prevalence of *G. duodenalis* among pre-weaned calves with diarrhea according to age.**

| Age (days) | No. of samples | No. of positive samples | $\chi^2$ (*P*-value) |
|---|---|---|---|
| 1−10 | 219 | 3 (1.4%) | 9.417 (0.009) |
| 11−30 | 186 | 14 (7.5%) | |
| 31−60 | 50 | 3 (6.0%) | |
| Total | 455 | 20 (4.4%) | |

## Subtypes of assemblages A and E

From the 20 samples that were positive for *bg*, 16, 5, and 6 sequences were obtained following genotyping of *bg*, *gdh*, and *tpi*, respectively. The sequence analysis of these genes revealed that *G. duodenalis* in our samples consisted of assemblages A and E (Table 4); this result was confirmed by phylogenetic tree of each gene (Figs 1–3). Assemblage E was more prevalent than assemblage A in pre-weaned calves with diarrhea. Moreover, the distribution of subtypes varied by region, with the E3 subtype detected in three regions (Table 2). Sequence analysis of the *bg* locus showed the presence of assemblage E (*n* = 15) and A (*n* = 1) in the samples. Further subtype analysis revealed that assemblage A was classified into sub-assemblage AI, whereas assemblage E were divided into the four subtypes, E1 (*n* = 3), E3 (*n* = 5), E5 (*n* = 3), and E11 (*n* = 4) (Table 5). We detected no novel *bg* subtype, and the subtypes detected were identical to those reported in a previous study [27]. Sequencing analysis at the *gdh* locus revealed the presence of five subtypes: of these, AI (*n* = 1), E1 (*n* = 1), and E12 (*n* = 1) were previously known,

**Table 4. Multilocus genotyping of *G. duodenalis* identified in this study based on three loci.**

| Sample ID | MLG | Genotype | | |
|---|---|---|---|---|
| | | *bg* | *gdh* | *tpi* |
| JB7_33d* | MLG-E1 | E5 | E12 | E57[a] |
| JB10_33d | MLG-E2 | E5 | E1 | E58[a] |
| JB34_30d | | E3 | – | + |
| JB12_15d | | E3 | – | – |
| GB1_30d | | E3 | – | E24 |
| JB16_3d | | E1 | – | – |
| JB3_30d | | E1 | – | E11 |
| JB6_10d | | E3 | N/A | N/A |
| JB4_30d | | E1 | + | – |
| GN25_13d | | E11 | E45[b] | + |
| GN22_13d | | E11 | – | + |
| GN24_17d (Mixed A+E) | | E11 | – | AI |
| GN27_20d | | AI | AI | + |
| GN53_12d (Mixed A+E) | | E3 | – | AI |
| GN56_11d | | E11 | E46[b] | + |
| JB2_22d | | E5 | + | – |
| Sequenced samples | | 16 | 5 | 6 |

"+": PCR positive but sequencing is not good.

"–": not detected.

"N/A": not tested.

"*": day.

"[a, b]": novel subtype.

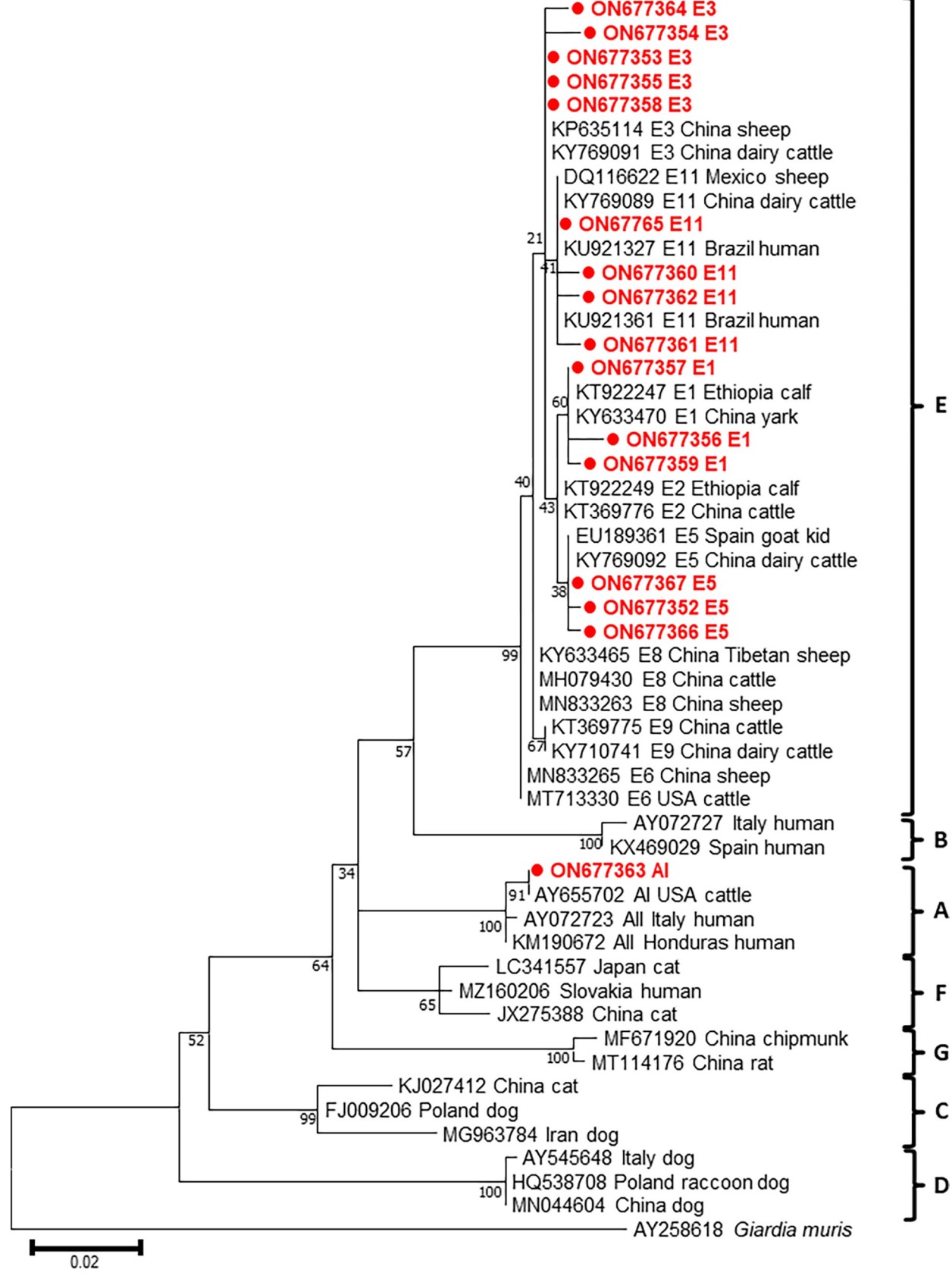

**Fig 1. Phylogenetic analysis based on *bg* of *Giardia duodenalis* using the Maximum-likelihood method with Tamura-Nei (TN93) model.** The numbers over branches indicate bootstrap values as a percentage of 1000 replicates that support each phylogenetic branch. Only bootstrap values >70% are shown. *G. duodenalis* sequences identified in this study are indicated by a bold circle symbol.

and E45 (*n* = 1) and E46 (*n* = 1) were identified as two novel subtypes (Table 6). Based on the analysis of the *tpi* locus, two and four sequences belonging to assemblages A and E, respectively, were identified. These assemblages consisted of three previously reported subtypes, namely AI (*n* = 2), E11 (*n* = 1), and E24 (*n* = 1). The remaining two sequences were novel subtypes, named as E57 (*n* = 1) and E58 (*n* = 1) (Table 7). Overall, our results revealed a higher prevalence of assemblage E than assemblage A in pre-weaned calves.

## Multilocus genotypes of *G. duodenalis*

Of the 16 samples, only two assemblage E-positive samples yielded amplicons from all three loci, and these were named MLG-E1 and MLG-E2. These two (MLG-E1 and MLG-E2) were found in calves aged 30 and 33 days, respectively, in Jeonbuk province (Table 4). Seven samples yielded amplicons from two loci (Table 4), with high degrees of genetic heterogeneity within each gene fragment. The *tpi* locus contained more polymorphic regions than the other two loci (Table 7). Interestingly, two samples showed discordant genotyping results among two loci (*bg* and *tpi*), indicating mixed infections involving assemblages A and E (Table 4). Mixed infections were found in 12- and 17-day-old calves from Gyeongnam province. This is the first study to report mixed infections in calves in the ROK.

## Discussion

This study showed that the prevalence of *G. duodenalis* was relatively low compared with the results of previous studies in the ROK [25, 27, 30–32]. Moreover, the infection rate of *G. duodenalis* was considerably lower than that reported globally [3, 13, 33, 34]. These variations may be attributed to the differences in geographical location, age, sample size, sampling period, management systems of farms, and detection method. Analysis of the SSU rRNA gene provides the highest sensitivity and has been commonly used for the detection of *G. duodenalis*; however, the sequence information obtained using this gene might be inadequate for the accurate identification of assemblages [3]. In contrast, analysis of *bg* is appropriate for the detection and multilocus genotyping of *G. duodenalis*. Most of all, the primary reason for the low prevalence of *G. duodenalis* in this study may be the implementation of improved management practices, such as the provision of clean water, disinfection, and hygiene (frequent removal of feces) in the farms involved in this study. In general, *Giardia* is associated with low hygiene conditions [35, 36]. Unlike farms using the old management systems, current livestock farms are larger and more specialized; thus, the management system in the ROK is highly focused on animal health.

The results of the present study revealed that the prevalence of *G. duodenalis* was different across regions, although its prevalence was not statistically significant. The highest infection rate of *G. duodenalis* was observed in Gyeongnam province in the southern part of the country. This may be because the occurrence of *G. duodenalis* is associated with climate, and due to climate change, the climate of the ROK has become increasingly hot and humid, particularly in the south. Cysts of *Giardia* are highly resistant to humid conditions [37, 38], and this may allow them to survive longer in the south [37–39]. It is possible that the high prevalence of *G. duodenalis* in this south is due to factors associated with climate; however, further studies are needed to evaluate the relationship between survival of *G. duodenalis* and environmental conditions.

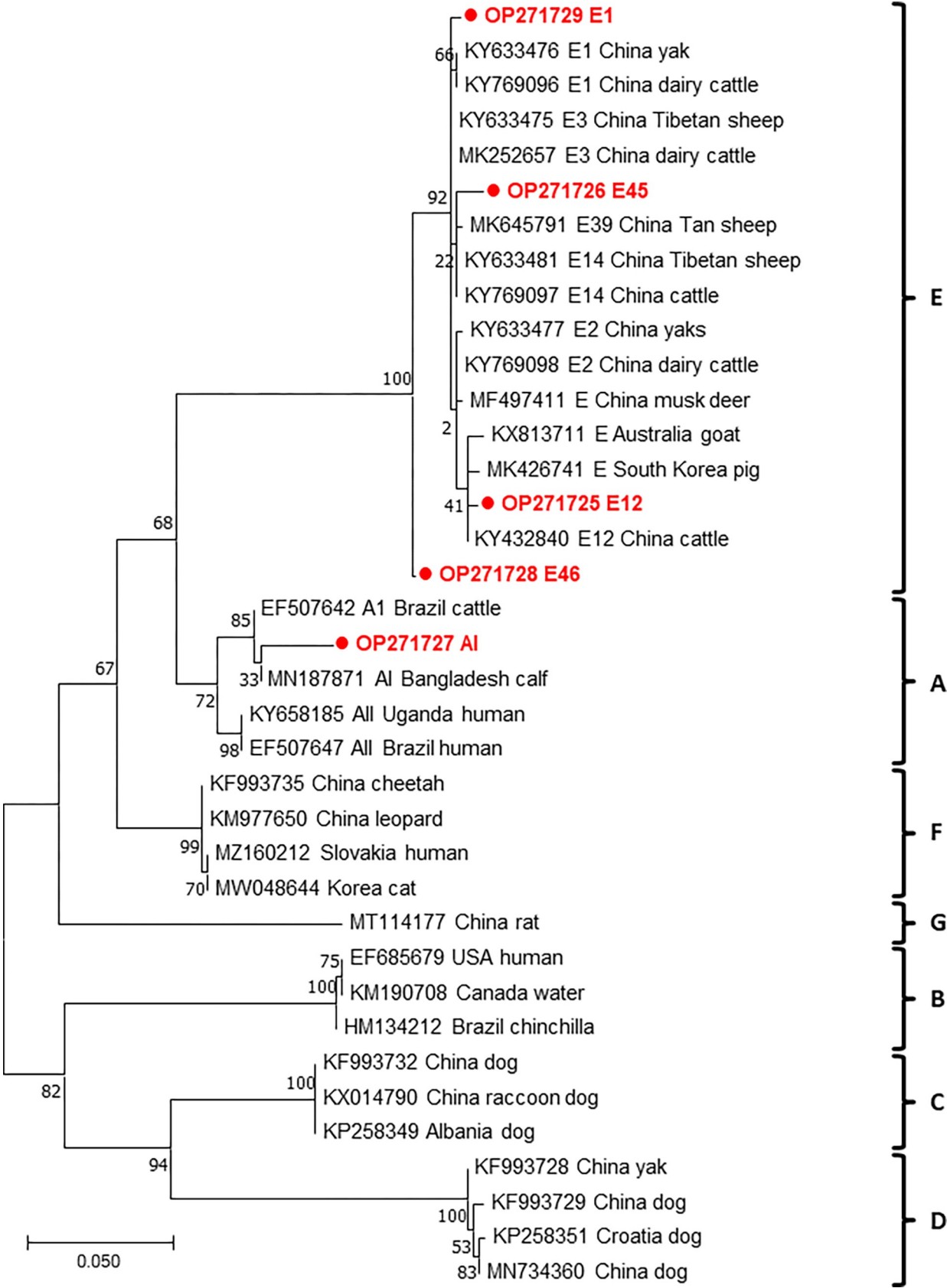

**Fig 2. Phylogenetic analysis based on *gdh* of *Giardia duodenalis* using the Maximum-likelihood method with Tamura-3-parameter (TN92) model.** The numbers over branches indicate bootstrap values as a percentage of 1000 replicates that support each phylogenetic branch. Only bootstrap values >70% are shown. *G. duodenalis* sequences identified in this study are indicated by a bold circle symbol.

The association between the prevalence of *G. duodenalis* and age of calves has been reported in several studies [27, 28, 40]. These reports indicate that *G. duodenalis* is relatively more prevalent in calves aged >1 month. However, our findings showed that the prevalence of *G. duodenalis* was the highest in calves aged 11–30 days. This finding is consistent with that of other studies that revealed significantly higher prevalence of *G. duodenalis* in young calves than in older animals [41–43]. Interestingly, the lowest infection rate of *G. duodenalis* was detected in neonatal calves and this result was consistent with our previous findings [27]. As the criteria for age classification in this study differed from that of a previous study [27], no firm conclusions could be drawn. However, our results strongly suggest that pre-weaned calves are at a higher risk of *G. duodenalis* infection than neonatal calves. The role of *G. duodenalis* as a primary cause of diarrhea remains controversial. Nevertheless, giardiasis is a condition that essentially leads to alterations in the microvilli, including a decreased crypt to villus ratio and brush border enzyme deficiencies [11], resulting in malabsorption, ill-thrift, and diarrhea. These results indicate that *G. duodenalis* should be considered as a causative agent of diarrhea, particularly in pre-weaned calves.

Our findings revealed that assemblage E is the most predominant in the ROK. This result is consistent with the findings of our previous study [28] and those of other studies [13, 41, 44, 45]. Currently, assemblage E is considered to have the potential of a zoonotic transmission. However, it has not yet been identified in humans in the ROK. Assemblage A, also identified in the present study, is capable of zoonotic transmission and is classified into three subtypes, AI, AII, and AIII, which have been mainly reported in livestock, humans, and wildlife, respectively [12]. Sub-assemblage AI is not only found in ruminants but can also infect humans. Interestingly, assemblage A is widespread in the United States cattle population [46], whereas it is relatively rare in calves in other countries [20, 35, 47, 48]. This suggests that different assemblages of *G. duodenalis* are circulating in different countries. In the present study, we identified four subtypes of assemblage E (E1, E3, E5, and E11) and this result is consistent with the findings of our previous study [27], indicating that these subtypes are prevalent in pre-weaned calves in the ROK. Cattle are known to be source of contamination of ground and surface waters, resulting in waterborne outbreak in humans [49–52]. Therefore, continuous monitoring in cattle is needed to prevent and control *G. duodenalis* infections in public health significance.

The phylogenetic analysis of *G. duodenalis* based on each gene revealed the presence of two distinct assemblages, A and E. The results of the present study demonstrated that *tpi* is a highly heterogenic genetic marker and can be used to differentiate *G. duodenalis* at the subtype level [53]. Moreover, a higher genetic diversity of *G. duodenalis* was observed within assemblage E. However, subtyping designation of assemblage E is not clearly supported by phylogenetic analyses. The difficulty in the classification of subtypes may be due to high inter- and intra-sequence variabilities in assemblage E [40, 54, 55]. In contrast to assemblage E, assemblage A showed low genetic diversity. Phylogenetic trees constructed herein provided data on grouping and genetic relationship but did not present accurate information on subtypes, particularly within assemblage E. These results suggest that multilocus sequencing is more useful for subtyping of assemblage E.

In this study, we report mixed infections for the first time. Assemblage-specific PCRs are used to detect mixed infections; however, these assays are less sensitivity, particularly in

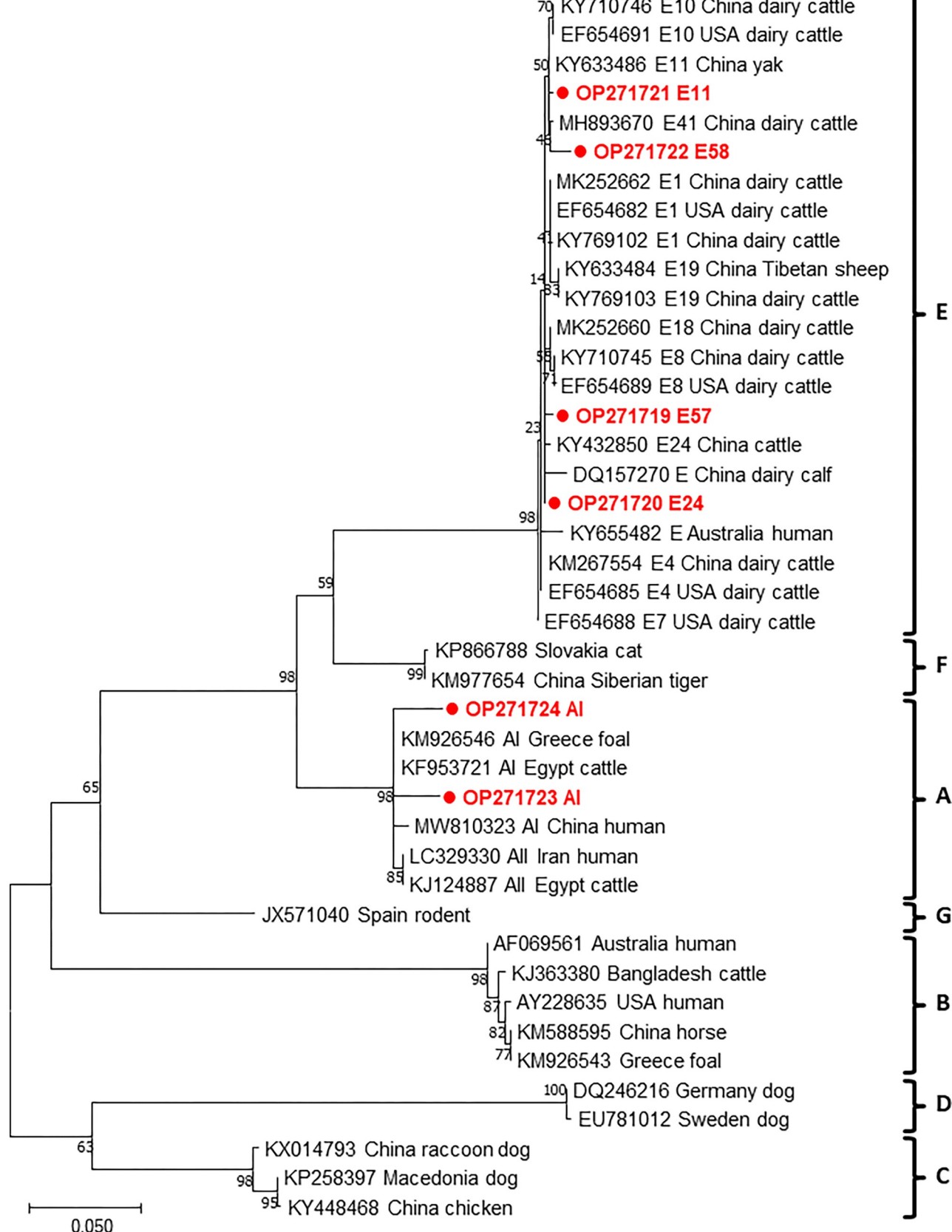

**Fig 3. Phylogenetic analysis based on _tpi_ of _Giardia duodenalis_ using the Maximum-likelihood method with Tamura-Nei (TN93) model.** The numbers over branches indicate bootstrap values as a percentage of 1000 replicates that support each phylogenetic branch. Only bootstrap values >70% are shown. _G. duodenalis_ sequences identified in this study are indicated by a bold circle symbol.

**Table 5. Sequence variations in the _bg_ locus of _G. duodenalis_ assemblage E among pre-weaned calves in the ROK.**

| Subtype | Sample | Nucleotide at position | | | |
|---|---|---|---|---|---|
| | | **117** | **222** | **324** | **465** |
| Ref. | DQ116624 | T | A | C | T |
| E1 | JB16_3d* | T | A | T | T |
| | JB3_30d* | T | A | T | T |
| | JB4_30d* | T | A | T | T |
| E3 | JB34_30d* | T | A | C | C |
| | JB12_15d* | T | A | C | C |
| | GB1_30d* | T | A | C | C |
| | JB6_10d* | T | A | C | C |
| | GN53_12d* | T | A | C | C |
| E5 | JB7_33d* | C | A | C | T |
| | JB2_22d* | C | A | C | T |
| | JB10_30d* | C | A | C | T |
| E11 | GN25_13d* | T | G | C | C |
| | GN22_13d* | T | G | C | C |
| | GN24_17d* | T | G | C | C |
| | GN56_11d* | T | G | C | C |

"*": sequences obtained in this study.

**Table 6. Sequence variations in the _gdh_ locus of _G. duodenalis_ assemblage E among pre-weaned calves in the ROK.**

| Subtype | Sample | Nucleotide at position | | | | | | | | | | |
|---|---|---|---|---|---|---|---|---|---|---|---|---|
| | | **271** | **277** | **418** | **440** | **454** | **463** | **508** | **538** | **589** | **591** | **608** |
| Ref. | KX813711 | G | A | G | C | G | T | T | T | T | G | A |
| E1 | JB10_30d* | G | A | T | C | G | T | T | T | T | G | G |
| E12 | JB7_33d* | G | G | G | C | G | T | T | T | T | G | G |
| E45[a] | GN25_13d* | G | G | T | C | A | T | T | T | T | A | G |
| E46[a] | GN56_11d* | A | A | T | A | G | C | C | C | A | G | G |

"*": sequences obtained in this study.
"a": novel subtype.

**Table 7. Sequence variations in the _tpi_ locus of _G. duodenalis_ assemblage E among pre-weaned calves in the ROK.**

| Subtype | Sample | Nucleotide at position | | | | | | | | | |
|---|---|---|---|---|---|---|---|---|---|---|---|
| | | **26** | **28** | **32** | **88** | **98** | **113** | **313** | **320** | **331** | **357** |
| Ref. | DQ157270 | T | T | C | T | C | C | T | G | G | C |
| E11 | JB3_30d* | C | C | G | T | T | C | T | G | A | C |
| E24 | GB1_30d* | C | C | G | T | C | C | T | G | A | C |
| E57[a] | JB7_33d* | C | C | G | T | C | C | T | A | A | C |
| E58[a] | JB10_30* | C | C | G | G | T | A | C | G | A | A |

"*": sequences obtained in this study.
"a": novel subtype.

detecting genetic variations, compared with genotyping based on single or multiple gene targets [56, 57]. Although next generation sequencing has been applied to accurately identify mixed infections, this method is extremely expensive. Mixed infections have been reported in Belgium [58], the UK [59], Germany [49], China [14], and the USA [46]. Compared with the findings in these countries, we discovered that mixed infections in calves were less common in the ROK. At this point, we are uncertain whether the mixed infections in this study were due to genetically different cysts or different alleles in the nuclei of a single cyst [60]. Furthermore, we did not evaluate the clinical signs of calves with mixed infections or the amount of cysts they shed into feces compared with those with either assemblage E or A infection. Mixed infections are of interest because infected calves may harbor the potentially zoonotic assemblage A and should therefore be appropriately diagnosed. Further investigations are required to determine the occurrence and pathogenicity of mixed infections.

The MLG approach is a reliable method to analyze the genetic diversity of *G. duodenalis* assemblages [61]. In this study, only two samples yielded amplicons of all three loci. Despite the small sample size, we detected genetic variations within assemblage E, implying that the level of subtype diversity found in this study was higher than that found in our previous study [27]. It is possible that the genetic heterogeneity within assemblage E is due to frequent intra-assemblage genetic recombination [29, 40, 54, 55]. The MLG shown in this study confirmed the presence of mixed assemblages in samples due to inconsistent assemblage designation at different genetic loci. This is evident as only two samples assigned as assemblage E at the *bg* locus were genotyped as sub-assemblage AI at the *tpi* locus. Therefore, MLG is a suitable method for detecting mixed infections and identifying potentially zoonotic assemblages in cattle and other hosts [14, 20, 62–64].

## Conclusions

In this study, the prevalence of *G. duodenalis* in pre-weaned calves with diarrhea was low. *G. duodenalis* infection was significantly associated with age of calves, with a relatively high prevalence in calves aged 11–30 days. Pre-weaned calves were at a higher risk of *G. duodenalis* infection than neonatal calves. Our findings indicate that *G. duodenalis* should be considered as a primary causative agent of diarrhea in pre-weaned calves. Both assemblages A and E were identified in calves, with assemblage E being the most prevalent. To the best of our knowledge, this is the first report describing mixed infections with assemblage A and E in calves in the ROK. These results suggest that calves are an important zoonotic reservoir and pose a potential risk for humans. Additional epidemiological studies are warranted to better understand the transmission and public health significance of *G. duodenalis* in the ROK.

## Acknowledgments

We thank Jeong-Byoung Chae, DVM for collecting feces.

## Author Contributions

**Conceptualization:** Jinho Park, Kyoung-Seong Choi.

**Data curation:** Kyoung-Seong Choi.

**Formal analysis:** Yu-Jin Park, Hyung-Chul Cho, Dong-Hun Jang, Jinho Park.

**Funding acquisition:** Kyoung-Seong Choi.

**Investigation:** Yu-Jin Park, Hyung-Chul Cho, Dong-Hun Jang.

**Supervision:** Kyoung-Seong Choi.

**Visualization:** Yu-Jin Park.

**Writing – original draft:** Yu-Jin Park, Kyoung-Seong Choi.

**Writing – review & editing:** Yu-Jin Park, Hyung-Chul Cho, Kyoung-Seong Choi.

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
