## [Decision Letter · Decision Letter 0]

31 Oct 2022

PONE-D-22-24630Multilocus genotyping of Giardia duodenalis in calves with diarrhea in the Republic of KoreaPLOS ONE

Dear Dr. Kyoung-Seong Choi,

Thank you for submitting your manuscript to PLOS ONE. After careful consideration, we feel that it has merit but does not fully meet PLOS ONE’s publication criteria as it currently stands. Therefore, we invite you to submit a revised version of the manuscript that addresses the points raised during the review process.

We look forward to receiving your revised manuscript.

Kind regards,

Saeed El-Ashram

Academic Editor

PLOS ONE

Journal Requirements:

"This research was supported by the National Research Foundation of Korea (NRF), which is funded by the Mid-Career Research Program (grant no. NRF-2021R1A2C1011579)."

"NO, The funders had no role in study design, data collection and analysis, decision to publish, or preparation of the manuscript."

Reviewers' comments:

Reviewer's Responses to Questions

**Comments to the Author**

1. Is the manuscript technically sound, and do the data support the conclusions?

Reviewer #1: Yes

Reviewer #2: Yes

2. Has the statistical analysis been performed appropriately and rigorously? 

Reviewer #1: Yes

Reviewer #2: Yes

3. Have the authors made all data underlying the findings in their manuscript fully available?

Reviewer #1: Yes

Reviewer #2: Yes

4. Is the manuscript presented in an intelligible fashion and written in standard English?

Reviewer #1: Yes

Reviewer #2: Yes

5. Review Comments to the Author

Reviewer #1: The current manuscript: Multilocus genotyping of Giardia duodenalis in calves with diarrhea in the Republic of Korea is well written and of interest. Meanwhile there are major points should be addressed.

1- Tables 1-4 were not included in the current version.

2- Please, include phylogenetic trees for the obtained sequences and discuss the obtained findings.

3- Author contributions, the revised version was not mentioned. All authors should approve the submitted version.

4- Double check on the cited references.

Reviewer #2: In this study entitled “Multilocus genotyping of Giardia duodenalis in calves with diarrhea in the Republic of Korea”, the authors investigated the infection of Giardia duodenalis in calves in ROK., and sequencing analyses showed that they belonged to zoonotic assemblage A. It was a meaningful survey of Giardia duodenalis in calves, but there were some questions.

Some questions:

1. I thought the title was vague and could be modified to better convey the message of the manuscript. Perhaps it would be appropriate to add "pre-weaned" to the title. For calves include pre weaned calves and post weaned calves.

2. Abstract: lines 22-25, “…causes diarrhea…”. The language of the paper should be concise, and the authors should avoid repeatitive description.

3. Introduction.

Based on the information about the Giardia duodenalis provided by the authors (also include the results of this article), I learned that Giardia duodenalis is an opportunistic parasitic. But in lines 51-53 (Giardia duodenalis is a …in extreme cases) and lines 63-64(Giardia duodenalis is a … industry), the author vaguely described that as long as the host is infected with Giardia, it can have symptoms. Whether I understand it incorrectly?

Lines 60-62. According to the references provided by the author, I found that the assemblage E have the potential zoonotic transmission, why does the author describe “indicating zoonotic transmission”? Please also check the whole text.

Lines 63-64 and lines73-79. Please add references and citations.

Line 85 “age group”? In the article, the author only investigated two age groups. Please give an accurate description.

4. Method.

The calves were housed together? or in a separate cage? Is the sampling season different? Whether seasonal information can be provided? More details about sample collection should be given in the Materials and methods section.

DNA extraction and PCR. Line 108. The author describes that the SSU rRNA gene locus has highest sensitivity, why not use SSU rRNA genes for screening? I suggest the author re-screen, otherwise the results of the survey may be wrong.

5. Discussion is too long and mostly repeats what were described in Results.

6. Lines 195-197 and lines 239-240 (In general … animal health.) and (cattle are known …). Please add references and citations.

Lines 266-270. Please add references and citations.

6. PLOS authors have the option to publish the peer review history of their article (what does this mean?). If published, this will include your full peer review and any attached files.

Reviewer #1: **Yes: **Waleed M. Arafa

Reviewer #2: No

<quillbot-extension-portal></quillbot-extension-portal>

---

## [Author Response · Author response to Decision Letter 0]

24 Nov 2022

We thank the reviewers for their careful reading and critical insights. These constructive comments were valuable and helpful for revising and improving our manuscript. We have provided detailed responses to the points raised by the reviewers and have modified our manuscript accordingly. The revised portions have been highlighted in red in the revised manuscript. We hope that the revised manuscript will be suitable for publication in PLoS One.

Reviewer #1: The current manuscript: Multilocus genotyping of Giardia duodenalis in calves with diarrhea in the Republic of Korea is well written and of interest. Meanwhile there are major points should be addressed.

1- Tables 1-4 were not included in the current version.

Response: We are sorry about that. Now, we have provided all tables in the manuscript.

2- Please, include phylogenetic trees for the obtained sequences and discuss the obtained findings.

Response: We agree with the reviewer’s comment. We have provided the phylogenetic trees. Please see figures. We have addressed the obtained findings in the discussion. Please see lines 293-303.

3- Author contributions, the revised version was not mentioned. All authors should approve the submitted version.

Response: All authors have approved the final version. We have mentioned regarding this. Please see lines 354-355.

4- Double check on the cited references.

Response: We agree with the reviewer’s comment. We made several mistakes for citation. We have provided all citations properly.

 

Reviewer #2: In this study entitled “Multilocus genotyping of Giardia duodenalis in calves with diarrhea in the Republic of Korea”, the authors investigated the infection of Giardia duodenalis in calves in ROK., and sequencing analyses showed that they belonged to zoonotic assemblage A. It was a meaningful survey of Giardia duodenalis in calves, but there were some questions.

Some questions:

1. I thought the title was vague and could be modified to better convey the message of the manuscript. Perhaps it would be appropriate to add "pre-weaned" to the title. For calves include pre weaned calves and post weaned calves.

Response: We have added in response to the reviewer’s comment. Please see the title.

2. Abstract: lines 22-25, “…causes diarrhea…”. The language of the paper should be concise, and the authors should avoid repeatitive description.

Response: We agree with the reviewer’s comment. We have revised these sentences in response to the reviewer’s comment. Please see lines 14-16. 

3. Introduction.

Based on the information about the Giardia duodenalis provided by the authors (also include the results of this article), I learned that Giardia duodenalis is an opportunistic parasitic. But in lines 51-53 (Giardia duodenalis is a …in extreme cases) and lines 63-64(Giardia duodenalis is a … industry), the author vaguely described that as long as the host is infected with Giardia, it can have symptoms. Whether I understand it incorrectly?

Response: We agree with the reviewer’s comment. We are sorry for the confusion. Giardia duodenalis infection can cause mild to severe symptoms in various hosts according to the papers we cited. In addition, Giardia duodenalis acts as a causative agent of diarrhea in calves. Please see lines 44-45 and lines 55-56.

Lines 60-62. According to the references provided by the author, I found that the assemblage E have the potential zoonotic transmission, why does the author describe “indicating zoonotic transmission”? Please also check the whole text.

Response: We have revised the sentence in response to the reviewer’s comment. Please see line 54.

Lines 63-64 and lines73-79. Please add references and citations.

Response: We have provided the references. Please see lines 56, 66, 67, and 71.

Line 85 “age group”? In the article, the author only investigated two age groups. Please give an accurate description.

Response: We agree with the reviewer’s comment. We are sorry for the confusion. We are divided into the three age groups (1-10 days, 11-30 days, and 31-60 days). Our aim is to identify the age group most associated with the occurrence of Giardia duodenalis. We have revised this to be precise according to the reviewer’s comment. Please see lines 77-78. An explanation of age group was added to Materials and methods. Please see lines 90-92.

4. Method.

The calves were housed together? or in a separate cage? Is the sampling season different? Whether seasonal information can be provided? More details about sample collection should be given in the Materials and methods section.

Response: We understand the reviewer’s concern. Unfortunately, we don’t know the exact information regarding rearing system. The veterinarian collected feces from individual diarrheic calf and sent to us with information such as gender, age, region, and sampling date. We can exactly know the seasonal information by the sampling date. We have provided this information. Please see lines 96-99. 

DNA extraction and PCR. Line 108. The author describes that the SSU rRNA gene locus has highest sensitivity, why not use SSU rRNA genes for screening? I suggest the author re-screen, otherwise the results of the survey may be wrong.

Response: We understand the reviewer’s concern. To be honest, we had not thought about detection of Giardia duodenalis using the SSU rRNA gene. This is a big mistake. We used the bg gene for the first screening. Most of all, for multilocus analysis, because three loci (bg, gdh, and tpi) are generally used, we did not screen Giardia duodenalis using the SSU rRNA gene. We think that analysis of bg is appropriate for the detection of Giardia duodenalis and in particular, this gives more information regarding the identification of subtype of assemblages.

5. Discussion is too long and mostly repeats what were described in Results.

Response: We agree with the reviewer’s comment. We have deleted the repetitive sentences as much as possible.

6. Lines 195-197 and lines 239-240 (In general … animal health.) and (cattle are known …). Please add references and citations.

Lines 266-270. Please add references and citations.

Response: We have provided the references in response to the reviewer’s comment. Please see lines 249, 290, 323, and 328.

---

## [Decision Letter · Decision Letter 1]

12 Dec 2022

Multilocus genotyping of Giardia duodenalis in pre-weaned calves with diarrhea in the Republic of Korea

PONE-D-22-24630R1

Dear Dr. Kyoung-Seong Choi,

We’re pleased to inform you that your manuscript has been judged scientifically suitable for publication and will be formally accepted for publication once it meets all outstanding technical requirements.

Kind regards,

Saeed El-Ashram

Academic Editor

PLOS ONE

Reviewers' comments:

Reviewer's Responses to Questions

**Comments to the Author**

1. If the authors have adequately addressed your comments raised in a previous round of review and you feel that this manuscript is now acceptable for publication, you may indicate that here to bypass the “Comments to the Author” section, enter your conflict of interest statement in the “Confidential to Editor” section, and submit your "Accept" recommendation.

Reviewer #1: All comments have been addressed

Reviewer #2: All comments have been addressed

2. Is the manuscript technically sound, and do the data support the conclusions?

Reviewer #1: Yes

Reviewer #2: Yes

3. Has the statistical analysis been performed appropriately and rigorously? 

Reviewer #1: Yes

Reviewer #2: Yes

4. Have the authors made all data underlying the findings in their manuscript fully available?

Reviewer #1: Yes

Reviewer #2: Yes

5. Is the manuscript presented in an intelligible fashion and written in standard English?

Reviewer #1: Yes

Reviewer #2: Yes

6. Review Comments to the Author

Reviewer #1: Authors responded properly to all the required points. I do not have more comments on the current version.

Thanks,

Reviewer #2: I check the full manuscript and response letter which was done by point to point , the authors already answered my questions, and revised them in the text.

7. PLOS authors have the option to publish the peer review history of their article (what does this mean?). If published, this will include your full peer review and any attached files.

Reviewer #1: No

Reviewer #2: No

<quillbot-extension-portal></quillbot-extension-portal>

---

## [Editor Report · Acceptance letter]

5 Jan 2023

PONE-D-22-24630R1 

Multilocus genotyping of *Giardia duodenalis* in pre-weaned calves with diarrhea in the Republic of Korea 

Dear Dr. Choi:

I'm pleased to inform you that your manuscript has been deemed suitable for publication in PLOS ONE. Congratulations! Your manuscript is now with our production department. 

Kind regards, 

on behalf of

Professor Saeed El-Ashram 

Academic Editor

PLOS ONE